# Transient marine euxinia at the end of the terminal Cryogenian glaciation

Xianguo Lang [1,2], Bing Shen[1], Yongbo Peng[3,4], Shuhai Xiao [5], Chuanming Zhou[6], Huiming Bao[1,3], Alan Jay Kaufman [7], Kangjun Huang[8], Peter W. Crockford[9,10,11], Yonggang Liu [12] Wenbo Tang[13] & Haoran Ma[1]

Termination of the terminal Cryogenian Marinoan snowball Earth glaciation (~650–635 Ma) is associated with the worldwide deposition of a cap carbonate. Modeling studies suggest that, during and immediately following deglaciation, the ocean may have experienced a rapid rise in pH and physical stratification followed by oceanic overturn. Testing these predictions requires the establishment of a high-resolution sequence of events within sedimentary records. Here we report the conspicuous occurrence of pyrite concretions in the topmost Nantuo Formation (South China) that was deposited in the Marinoan glacial deposits. Sedimentary facies and sulfur isotope data indicate pyrite precipitation in the sediments with $H_2S$ diffusing from the overlying sulfidic/euxinic seawater and Fe (II) from diamictite sediments. These observations suggest a transient but widespread presence of marine euxinia in an ocean characterized by redox stratification, high bioproductivity, and high-fluxes of sulfate from chemical weathering before the deposition of the cap carbonate.

[1] Key Laboratory of Orogenic Belts and Crustal Evolution, MOE, School of Earth and Space Science, Peking University, Beijing 100871, China. [2] State Key Laboratory of Palaeobiology and Stratigraphy, Nanjing Institute of Geology and Palaeontology and Center for Excellence in Life and Paleoenvironment, Chinese Academy of Sciences, Nanjing 210008, China. [3] Department of Geology and Geophysics, Louisiana State University, Baton Rouge, LA 70803, USA. [4] Shanghai Engineering Research Center of Hadal Science and Technology, College of Marine Sciences, Shanghai Ocean University, Shanghai 201306, China. [5] Department of Geosciences, Virginia Tech, Blacksburg, VA 24061, USA. [6] CAS Key Laboratory of Economic Stratigraphy and Paleogeography, Nanjing Institute of Geology and Palaeontology and Center for Excellence in Life and Paleoenvironment, Chinese Academy of Sciences, Nanjing 210008, China. [7] Department of Geology, University of Maryland, College Park, MD 20742, USA. [8] Shaanxi Key Laboratory of Early Life and Environments, Department of Geology, Northwest University, Xi'an 710069, China. [9] Department of Earth and Planetary Sciences, McGill University, Montreal H3A0E8, Canada. [10] Department of Earth and Planetary Sciences, Weizmann Institute of Science, Rehovot 76100, Israel. [11] Department of Geoscience, Princeton University, Princeton 08544, USA. [12] Department of Atmosphere and Ocean Sciences, School of Physics, Peking University, Beijing 100871, China. [13] School of Mathematical & Statistical Sciences, Arizona State University, Tempe, AZ 85287, USA. Correspondence and requests for materials should be addressed to B.S. (email: bingshen@pku.edu.cn) or to Y.P. (email: yongbopeng@gmail.com)

The Sturtian (~717–660 Ma) and Marinoan (~650–635 Ma) snowball Earth events in the Cryogenian Period[1–5] represent the most severe pan-glaciation climate experienced over the past 2 billion years, and possibly over the entire Earth history[2,6,7]. Climatic and geochemical models show that the termination of a snowball Earth glaciation requires a high level of atmospheric $CO_2$ ($pCO_2 > 0.2$ bar)[8–12] in order to overcome the high albedo from global ice cover. Such a high $pCO_2$ atmosphere will result in a rapid meltdown of glaciers[13] and an extreme greenhouse climate[9,10]. Rapid deglaciation may lead to the shutdown of thermohaline circulation and the intensification of ocean stratification[14,15], with warmer and dysoxic freshwater in the surface ocean overlying cold and anoxic seawater in the deep[13,15].

Such a catastrophic end of a snowball Earth glaciation is expected to drive drastic perturbations in ocean chemistry[16–19]. Globally, sedimentary sequences during deglaciation are mostly manifested as a cap carbonate sharply overlying glacial diamictites[2,7,20,21]. Although the cap carbonate is often assumed to have been deposited immediately after the onset of deglaciation, a window of time is required to allow atmospheric $pCO_2$ drawndown and seawater alkalinity buildup[9,20]. This necessarily means that deglaciation and associated continental weathering must have started before the initiation of cap carbonate deposition. This time lag is supported by recent work exploring Mg isotopes within post-Marinoan sequences[20] and may be on the order of $10^5$ years[9].

Because this time lag cannot be resolved using currently available geochronometric tools, we are forced to carry out high-resolution analysis of sedimentary sequence in order to test models about the termination of snowball Earth glaciation[22]. Here, we report the occurrence of abundant pyrite concretions from the topmost Nantuo Formation[5,23,24], a sedimentary sequence deposited during the terminal Cryogenian Marinoan glaciation in the Yangtze Block of South China (Fig. 1 and Supplementary Fig. 1, see Supplementary Note 1). These pyrite concretions lie immediately beneath the cap carbonate of the basal Doushantuo Formation deposited in the Ediacaran Period, and they present opportunities to investigate the atmospheric, oceanic, and biological events during the termination of the Marinoan glaciation.

In this paper, we explore the origin of these pyrite concretions based on an integrated analysis of sedimentary facies, thin-section petrography, and multiple sulfur stable isotopes. We show that there was a transient marine euxinia before the deposition of the cap carbonate.

## Results

**Stratigraphic distribution and petrography.** Pyrite concretions are abundantly distributed in the top 0.5–10 m of the Nantuo Formation (Fig. 2). The concretions include spheroidal-ellipsoidal nodules and aggregates with irregular outlines (Supplementary Fig. 2a–g). The pyrite nodules are randomly distributed and aligned parallel or subparallel to the bedding plane, whereas the pyrite aggregates are parallel to the bedding plane (Supplementary Fig. 2c). Individual concretions are isolated from each other, and there is no connection between nodules or between a nodule and an aggregate.

The pyrite concretions are present in all studied sections except those in proximal inner shelf environment of the Yangtze Block (Figs. 1, 2). Both the size and abundance of pyrite concretions display depth gradients. In the basinal sections, most pyrite nodules are 10–30 cm in size (Supplementary Fig. 2), whereas pyrite aggregates can be >1 m in length and >30 cm in thickness (Supplementary Fig. 2c). In these settings, pyrite concretions account for ~5 vol% (ranging from 2 to 10 vol%, $n = 5$, Supplementary Table 1) of the concretion-bearing strata. In the slope depositional environment, most pyrite nodules vary between 5 and 15 cm in size (Supplementary Fig. 2f), and pyrite concretions account for 2~5 vol% (average value of 3.1 vol%, $n = 5$, Supplementary Table 1). In the shallower outer shelf sections, pyrite concretions occur only sporadically (<0.5 vol%) as small nodules less than 3 cm in size in gravelly siltstone layers immediately below the Doushantuo cap carbonate (Supplementary Fig. 2g). No pyrite concretions are found in the most proximal inner shelf sections (Fig. 1).

Both the nodules and aggregates are composed of euhedral pyrite, which normally range from 50 to 500 μm with occasional occurrences of mega-crystals >1 mm in size (Supplementary Fig. 3a–d). No framboidal pyrite has been identified using reflective microscopy (Supplementary Fig. 3d), nor are

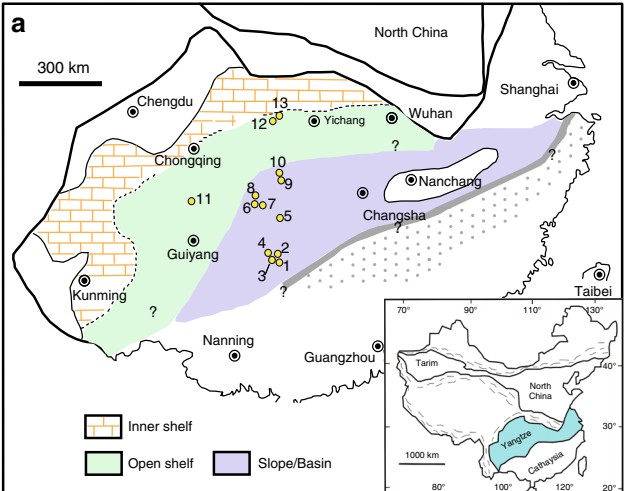

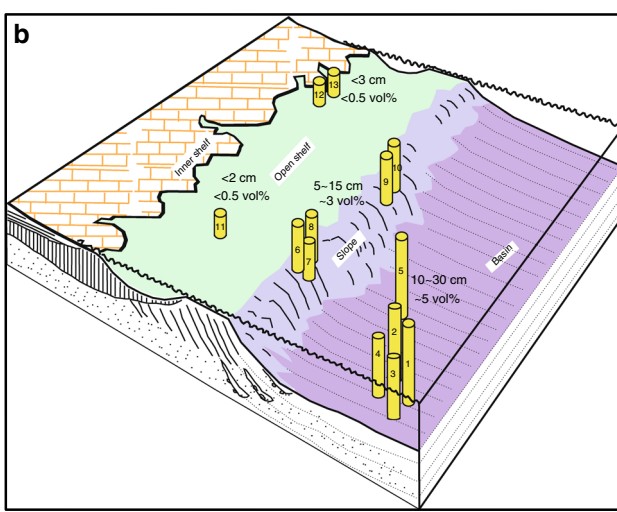

**Fig. 1** Paleogeographic map and depositional model of the Yangtze Block. **a** Paleogeographic map, modified from Jiang et al.[21] showing the distribution of pyrite concretions in the topmost Nantuo Formation. Inset showing the geographic locality of the Yangtze Block. **b** Depositional model. The height of the columns indicates the maximum size of pyrite nodules observed in field. 1: Yazhai, 2: Tongle, 3: Silikou, 4: Yangxi, 5: Yuanjia, 6: Huakoushan, 7: Bahuang, 8: Siduping, 9: Tianping, 10: Songlin, 11: Youxi, 12: Huajipo, 13: Shennongjia

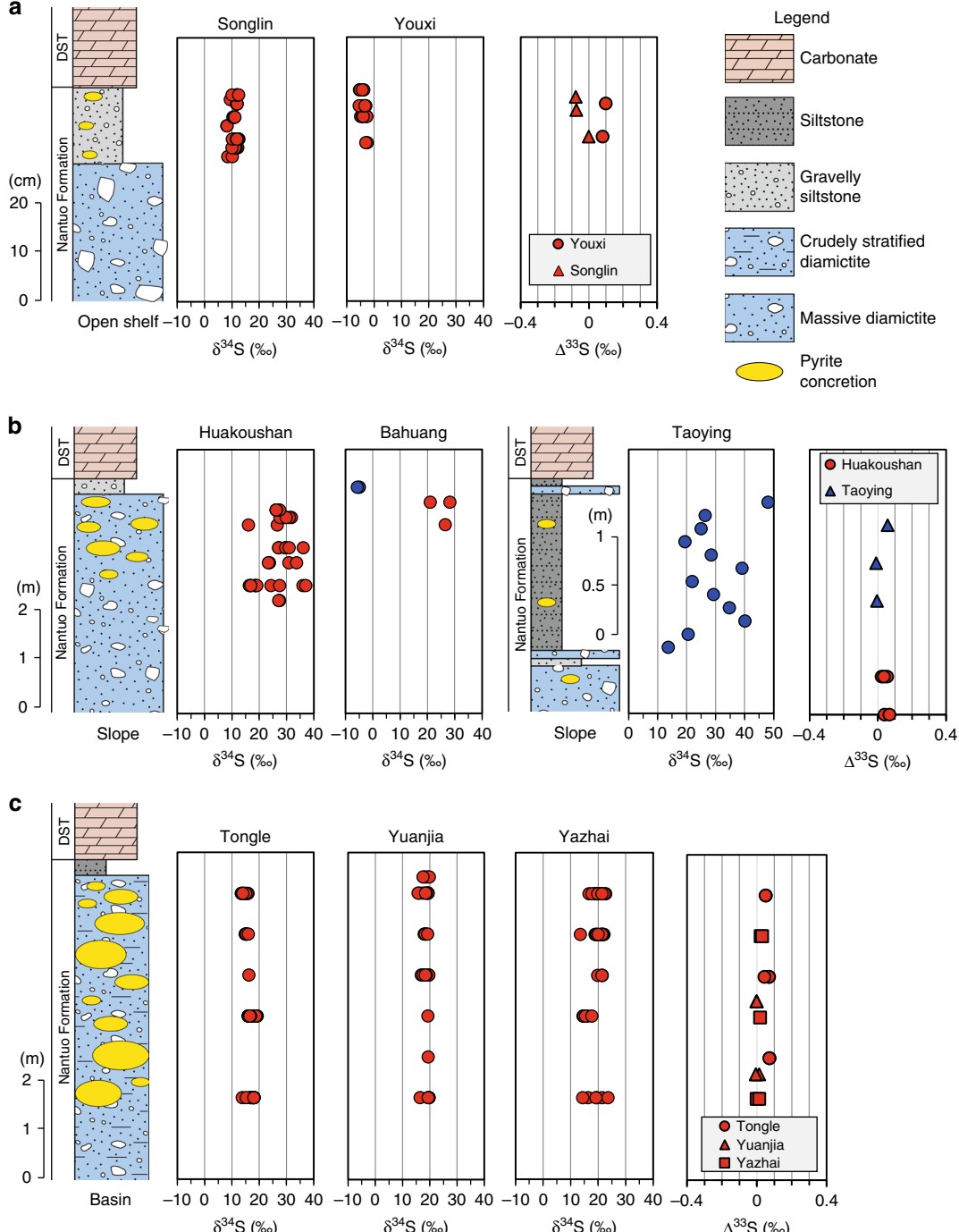

**Fig. 2** Sulfur isotopic compositions of pyrite concretions (red symbols) and disseminated pyrite (blue symbols) in the Nantuo Formation. **a** Limited variations of $\delta^{34}S_{py}$ and variable $\Delta^{33}S$ values of pyrite concretions from open shelf environment. **b** Variable $\delta^{34}S_{py}$ and slightly positive $\Delta^{33}S$ values of pyrite concretions and disseminated pyrite from slope environment. **c** Limited variations of $\delta^{34}S_{py}$ and variable $\Delta^{33}S$ values of pyrite concretions from basin environment. DST: Doushantuo Formation

framboidal cores present in euhedral pyrite under back-scatter electron microscopy (Supplementary Fig. 3e, f), suggesting that the euhedral crystals were not derived from overgrowths of framboidal pyrite. Pyrite nodules are composed of densely packed euhedral pyrite and interstitial space is filled with fine sands/silts and/or cemented by silica (Supplementary Fig. 3a–d). Within a single nodule, there is no rim to core differentiation in either pyrite content or crystal size. On the other hand, pyrite aggregates are texturally supported by siliciclasts that are identical to the host rock in composition, and thus have less (15–30 vol%) pyrite

content than the nodules (40–60 vol%). Pyrite crystals may contain abundant siliciclastic inclusions (Supplementary Fig. 3), and small pebbles up to 2 cm in size are observed in some aggregates and large nodules (Supplementary Fig. 2a and d).

**Sulfur isotopic compositions.** Individual pyrite concretions have limited variations in sulfur isotope composition ($\delta^{34}S_{py}$), but $\delta^{34}S_{py}$ may vary substantially between different concretions or between different sections. In basinal sections, concretions display a smaller range of variation in $\delta^{34}S_{py}$ [Tongle

(13.5–19.3‰, mean = 16.5‰), Yuanjia (15.7–19.9‰, 18.5‰), Yazhai (13.6–23.7‰, mean = 19.4‰)]. Concretions from the slope sections have higher $\delta^{34}S_{py}$ than those from the basinal sections, and $\delta^{34}S_{py}$ values display a wider range of variation [Huakoushan (16.1–37.1‰, mean = 26.7‰) and Bahuang (21.0–28.2‰, mean = 25.2‰)]. $\delta^{34}S$ values of disseminated pyrite extracted from gravelly siltstone in the top Nantuo Formation shows greater fluctuation (13.7–48.1‰, mean = 28.9‰) in the Taoying section but limited variation (–5.0‰ to –5.7‰, mean = –5.4‰) in the Bahuang section. In the outer shelf sections, pyrite concretions from the gravelly siltstone layers have the lowest $\delta^{34}S_{py}$ values [Songlin (8.1–12.6‰, mean = 11.1‰), Youxi (–5.5‰ to –2.5‰, mean = –4.0‰)]. $\Delta^{33}S$ (defined as $\Delta^{33}S = \delta^{33}S - 1000([1 + \delta^{34}S/1000]^{0.515} - 1))$ also exhibits a wide but overlapping range of variation among different concretions or among different facies [basin (–0.008‰ to 0.072‰, mean = 0.027‰), slope (–0.010‰ to 0.067‰, mean = 0.032‰), shelf (0.018‰ to 0.098‰, mean = 0.057‰)] (Fig. 2, Supplementary Tables 2, 3).

## Discussion

Pyrite can be generated by hydrothermal activities during late-stage diagenesis, precipitated directly from seawater column, or formed within sediment porewater during early diagenesis. Postdepositional hydrothermal origin of the Nantuo pyrite concretions can be ruled out on the basis of the following sedimentological, petrological, and geochemical observations. First, the stratigraphic distribution of pyrite concretions is controlled by depositional environment (Fig. 1). Pyrite concretions can be observed both in gravelly siltstone and massive diamictite in the slope environment, whereas in the shallow water facies pyrite nodules only occur in gravelly siltstone. Second, no fluid conduits or hydrothermal veins are observed in connection with pyrite concretions (Supplementary Fig. 2). Third, although the diamictite-cap carbonate interface could serve as a conduit for fluid circulation, pyrite nodules also occasionally occur in the overlying cap carbonate (Supplementary Fig. 2e). Pyrite nodules are preserved in the matrix of cap carbonate and predating the early diagenetic sheet crack structures and arguing against a late diagenetic or hydrothermal origin. Finally, highly positive $\delta^{34}S_{py}$ (>10‰) and variable $\Delta^{33}S$ values are not characteristics of hydrothermal pyrite[25–27] (Fig. 2). Direct pyrite precipitation from seawater is also inconsistent with observed euhedral pyrite (Supplementary Fig. 2), because sedimentary pyrite formed in the water column is typically framboidal in morphology[28]. Although euhedral pyrite could be generated by the overgrowth of framboidal pyrite[29], no framboidal cores have been identified (Supplementary Fig. 3e, f).

Petrological evidence suggest that the Nantuo pyrite concretions were formed in sediment during early diagenesis. Pyrite crystals are tightly packed with interstitial spaces filled with clasts or cemented by silica, or float within a siliciclastic matrix, suggesting early concretion formation before sediment compaction (Supplementary Figs. 2d and 3a, b). Furthermore, the matrix of pyrite-bearing diamictite and gravelly siltstone also contains some disseminated euhedral pyrite, (Supplementary Fig. 3g, h), suggesting an authigenic origin. Thus, the euhedral pyrite in the Nantuo concretions most likely precipitated from porewater within sediment.

In modern nonsulfidic oceans, authigenic pyrite precipitation is fueled by dissimilatory sulfate reduction (DSR) in sediment porewater, and DSR is sustained by sulfate diffusion from seawater[30] (Supplementary Fig. 4a). Alternatively, DSR may occur in a sulfidic water column, and authigenic pyrite can precipitate from porewater within sediment, with $H_2S$ diffusing from the overlying euxinic seawater (Supplementary Fig. 4b). In order to

differentiate these two scenarios, we develop numerical models to simulate the sulfur isotope systematics of pyrite formation. In the first scenario, DSR in sediment porewater is sustained by continuous diffusion of seawater sulfate, which is driven by a sulfate concentration gradient that results from porewater sulfate consumption by DSR. Here we simulate the processes utilizing a one-dimensional diffusion-advection-reaction (1D-DAR) model. Assuming a seawater sulfate sulfur isotopic composition ($\delta^{34}S_{sw}$) of +30‰[31] and biological fractionation ($\Delta_{DSR}$) in DSR at 40‰[32], our modeling results indicate a maximum $\delta^{34}S_{py}$ value of +8‰ (Fig. 3a, b), and the amount of pyrite formation is variably controlled by sedimentation rate, reaction rate of DSR, and seawater sulfate concentrations (Fig. 3c, Supplementary Figs. 5–9, and Supplementary Table 4). Therefore, DSR in sediment porewater cannot explain the high $\delta^{34}S_{py}$ values of the pyrite concretions in slope and basinal sections (Fig. 2).

Heavy ($^{34}S$-enriched) pyrite could be generated by DSR in a closed porewater system (i.e., without sulfate supply from seawater), which can be simulated by a Rayleigh distillation model (see Supplementary Note 2 and Supplementary Table 5). However, the Rayleigh process can only account for <0.1 vol% of pyrite (Fig. 3a), and thus cannot explain the high pyrite content in the top Nantuo Formation (Fig. 3a, Supplementary Table 1). On the other hand, pyrite precipitation in porewater, driven by $H_2S$ diffusion from sulfidic seawater, can be simulated by a combination of DSR in the water column and $H_2S$ diffusion/pyrite formation in sediments. We simulated these processes using a Rayleigh distillation model and a 1D-DR model (see Supplementary Note 2 and Supplementary Table 6). Our modeling result indicates that $\delta^{34}S_{py}$ values are predominantly controlled by DSR with high $\delta^{34}S_{py}$ values resulting from a high degree of DSR in the water column (Fig. 3d, e). In contrast, $H_2S$ diffusion controls the pyrite content but not $\delta^{34}S_{py}$ value (Supplementary Fig. 10), because the reaction between $H_2S$ and reactive Fe to generate pyrite is associated with negligible isotopic fractionation (~1‰)[33]. Our modeling results indicate that precipitation of $^{34}S$-enriched pyrite in the basin and slope sections was driven by a higher degree of sulfate reduction than in outer shelf settings (Fig. 3d, e).

Because $\delta^{34}S_{py}$ mirrors the isotopic composition of $H_2S$ in seawater[33], the spatial variation in $\delta^{34}S_{py}$ reflects the isotopic heterogeneity of seawater $H_2S$. The relatively consistent $\delta^{34}S_{py}$ values observed in the basinal samples suggest that the $H_2S$ concentration was more or less homogenous in most distal seawaters. For the same reason, the more variable $\delta^{34}S_{py}$ values of the slope sections reflect a rapid oscillation of seawater $H_2S$ concentrations.

Given high reactive Fe contents of siliciclastic sediments[34–36], the amount of pyrite formation was controlled by the availability of $H_2S$ in seawater, which was a function of seawater $H_2S$ concentrations and the volume of seawater. Thus, abundant pyrite in the top Nantuo Formation implies vigorous DSR and a high concentration of dissolved $H_2S$ in the seawater, which is also consistent with the lack of bacterial sulfate disproportionation as shown in the $\Delta^{33}S$–$\delta^{34}S$ plot (Fig. 3f)[37]. The decrease of pyrite content from deep to shallow settings is attributed to the bathymetric differences of the depositional environments[38]. Deeper water in the basin may have a thicker sulfidic water layer, resulting in more pyrite precipitation. While the absence of pyrite concretions in the inner shelf setting is the consequence of an absence of sulfidic seawater in the near shore, shallow, and more oxic waters[39,40].

Abundant pyrite concretions in the top Nantuo Formation imply the development of oceanic euxinia before the precipitation of cap carbonate[41]. Oceanic euxinia can only be sustained by sufficient supplies of sulfate and organic

matter[36,42,43]. Riverine influx is the major source of nutrients (e.g., P) and seawater sulfate[30,44,45]. We hypothesize that, during the deglaciation of the Marinoan glaciation, enhanced continental weathering[20] could have delivered abundant nutrients and sulfate into the ocean. Because the deglacial

continental weathering may predate oceanic euxinia[20] due to the delayed recovery of productivity in acidified seawater, both nutrients and sulfate might be accumulated in seawater from riverine inputs. Once surface ocean productivity was resumed at high levels, high nutrient content could sustain high

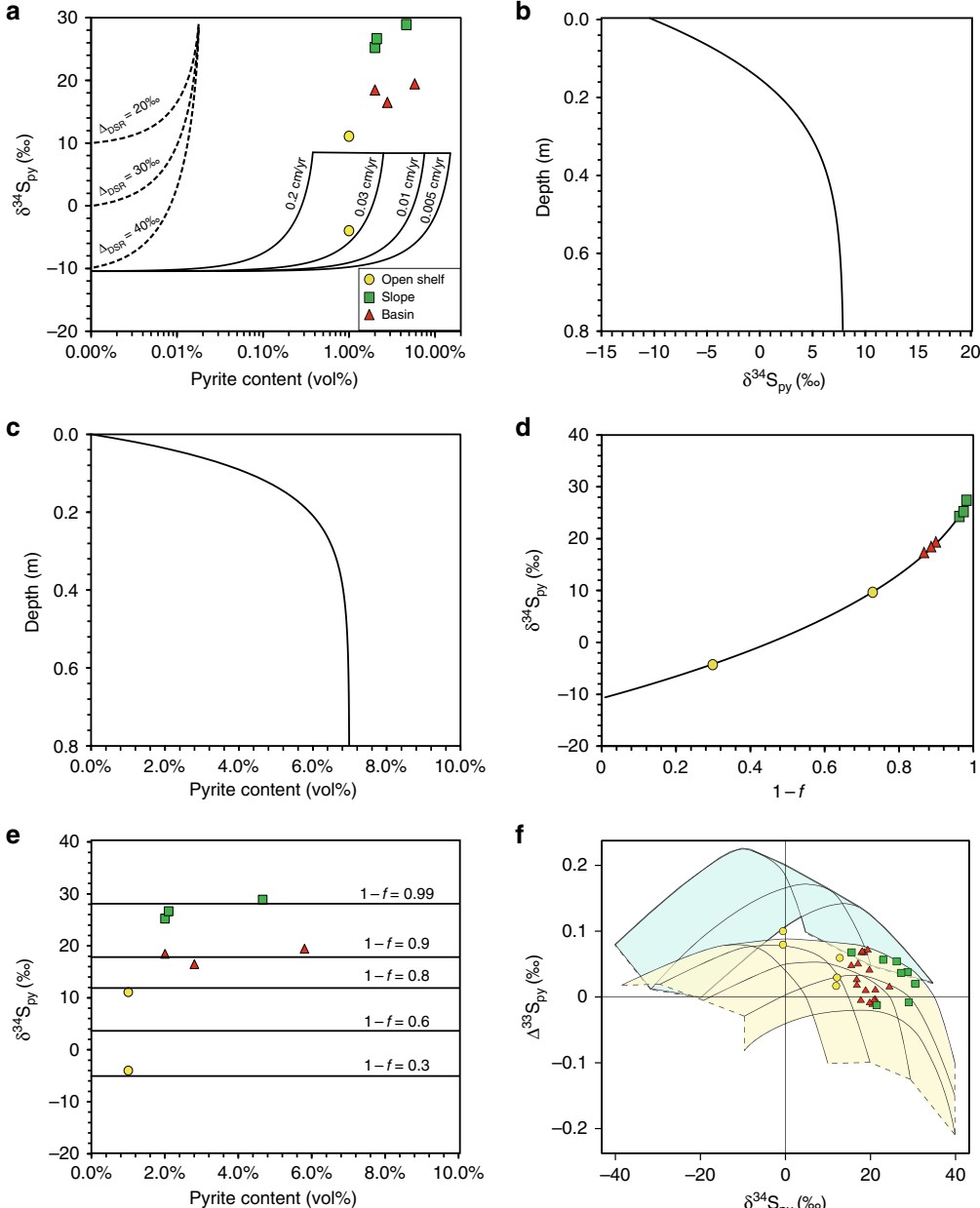

**Fig. 3** Modeling results. **a** The Rayleigh distillation model (dashed lines, with different sulfur isotope fractionations or $\Delta_{DSR}$) showing the relationship between $\delta^{34}S_{py}$ and pyrite content with DSR occurring within sediment porewater under a closed system. The 1D-DAR model result (solid lines, with different sedimentation rates) showing the relationship between $\delta^{34}S_{py}$ and pyrite content when DSR occurs in sediment porewater with sulfate supply by diffusion from the overlying seawater (an open system). **b** and **c** The maximum values of $\delta^{34}S_{py}$ and pyrite content are ~+8‰ and 6.99 vol% within sediments. The default parameters for $D_s$, $s$, $R$, and $[SO_4]_0$ are $3.61 \times 10^{-6}$ cm$^2$ s$^{-1}$, 0.01 cm year$^{-1}$, 1 year$^{-1}$, and 3 mM L$^{-1}$, respectively. **d** The Rayleigh distillation model quantifying the $\delta^{34}S$ of H$_2$S with DSR occurring in seawater. Assuming $\delta^{34}S_{sw}$ is 30‰, sulfur isotope fractionation for DSR is 40‰, variable $\delta^{34}S$ values of the Nantuo pyrite concretions indicates the different degree of DSR ($1 - f$, $f$ is the fraction of sulfate remaining). **e** The 1D-DR model result (black lines) showing DSR in water column followed by pyrite formation in sediment porewater fueled by H$_2$S diffusion from sulfidic seawater. This process can explain high $\delta^{34}S_{py}$ and high pyrite content in the basin and slope sections, indicating high degree of sulfate reduction in water column. **f** $\delta^{34}S_{py}$–$\Delta^{33}S_{py}$ cross-plot[37]. Individual contour lines represent modeled sulfur isotopic compositions of pyrite formed from a sulfate pool with an initial sulfur isotope at the right end of the lines. Modeled values in the blue field require a bacterial sulfur disproportion (BSD), whereas the yellow field indicate pyrite formed by DSR only. Measured data from pyrite in the top Nantuo Formation fall in the yellow field, suggesting that the pyrite was precipitated in an anoxic environment where DSR but not BSD occurs

productivity, while DSR in water column would be fueled by sufficient supplies of organic matter and sulfate, driving oceanic euxinia.

The pyrite-rich interval in the topmost Nantuo Formation marks a brief yet widespread euxinic condition in the aftermath of the Marinoan glaciation immediately before cap carbonate precipitation. It is an interval with rising sulfate levels[17], and high nutrient fluxes from the continents. This is also an interval characterized by redox stratified oceans where a large quantity of $H_2S$ was scavenged by Fe(II) in the glacial diamictite sediments. Importantly, this is also an interval prior to the deposition of the well-known cap carbonates, a time when seawater pH values or alkalinity were still below the threshold of carbonate precipitation or $pCO_2$ levels were sufficiently high in the atmosphere[12,46]. The current study extends previous efforts to reconstruct from sedimentary records the sequence of events in the aftermath of the Marinoan meltdown[47] in order to test the predictions of snowball Earth glaciations[2].

## Methods

**Sample preparation and sulfur isotope analysis.** Pyrite concretions were first split by a rock saw and then slabs were polished before sampling. Pyrite samples were collected by a hand-held micromill from the polished slabs. To capture the isotopic variations within a single pyrite concretion, multiple samples were sequentially collected from rim to core. Sulfur isotopic compositions of pyrite were measured at the Oxy-Anion Stable Isotope Consortium in Louisiana State University, at the EPS Stable Isotope Laboratory in McGill University, and at the Geochemistry laboratory in University of Maryland.

For $\delta^{34}S$ analysis, about 0.05 mg pyrite-rich powder was mixed with 1–2 mg $V_2O_5$, and was analyzed for S isotopic compositions on an Isoprime 100 gas source mass spectrometer coupled with a Vario Microcube Elemental Analyzer. Sulfur isotopic compositions are expressed in standard δ-notation as permil (‰) deviations from the Vienna-Canyon Diablo Troilite standard. The analytical error is <0.2‰ based on replicate analyses of samples and laboratory standards. Samples were calibrated on two internal standards: LSU-Ag₂S-1: −4.3‰; LSU-Ag₂S-2: +20.2‰.

Multiple sulfur isotope analyses were conducted by converting pyrite samples to $H_2S_{(g)}$ via the chromium reduction method. $H_2S$ gas was then carried through a $N_2$ gas stream to a Zn acetate solution where it precipitated as ZnS. ZnS was then converted to $Ag_2S$ through addition of $AgNO_3$. Samples were then filtered and dried at 80 °C. $Ag_2S$ samples were converted to $SF_6(g)$ through reaction with $F_2(g)$ in heated Ni bombs. Generated gas was then purified through a series of cryo-focusing steps followed by gas chromatography. The purified samples were then analyzed on a Thermo MAT-253 on dual-inlet mode. The total error on the entire analytical procedure is less than ($1\sigma$) 0.1‰ for $\delta^{34}S$ and 0.01‰ for $\Delta^{33}S$.

**Geochemical models of pyrite formation.** Sedimentary or authigenic pyrite can precipitate either from sediment porewater or water column. In sediment porewater, pyrite formation can occur in a closed or an open porewater system. For a closed system, DSR occurs within sediment porewater with no additional sulfate supply from seawater. In contrast, open system DSR refers to sulfate supply via diffusion from the overlying seawater and pyrite formation in sediment porewater (Supplementary Fig. 4).

The close system reaction can be simulated by the Rayleigh distillation model, whereas the open system pyrite formation can be modeled by the 1D-DAR or the 1D-DR model.

Rayleigh distillation model. The Rayleigh distillation model can be expressed as follows:

$$\delta^{34}S_r = \delta^{34}S_{sw} - 1000 \times \left( f^{(\alpha-1)} - 1 \right) \tag{1}$$

where $\delta^{34}S_{sw}$ is the isotopic composition of seawater, $\delta^{34}S_r$ is the isotopic composition of remaining sulfate in sediment porewater or in water column. $f$ is the fraction of sulfate remaining, and $\alpha$ is the fractionation factor for DSR. The isotopic composition of $H_2S$ ($\delta^{34}S_{H2S}$ or pyrite ($\delta^{34}S_{Py}$, assuming negligible fractionation between $H_2S$ and pyrite) can be described as:

$$\delta^{34}S_{py} = \frac{\left( \delta^{34}S_{sw} - \delta^{34}S_r \times f \right)}{1 - f} \tag{2}$$

The amount of pyrite ($M_{py}$) formation can be calculated by:

$$M_{py} = \frac{\emptyset \times (1-f) \times [SO_4] \times M_r}{2 \times \rho_{py} \times (1-\emptyset) \times 1000} \tag{3}$$

where ∅ is the porosity of sediment (estimated at 60%), $\rho_{py}$ is the density of pyrite (4.9 g cm⁻³), $M_r$ is the relative molecular mass of pyrite, $[SO_4]$ is seawater/porewater sulfate concentration (mol L⁻¹).

1D-DAR model. The 1D-DAR model can be applied to quantify the geochemical profiles in sediment porewater. The DAR model describes three physio-chemical processes: molecular diffusion, advection, and chemical reaction. During DSR in porewater, sulfate is supplied from seawater by diffusion. The diffusion is driven by the sulfate concentration gradient between seawater and sediment porewater, which is generated by sulfate consumption in sediment porewater by DSR. To model sulfur isotopic profiles of porewater, we treat ³⁴S and ³²S separately. The 1D-DAR model is expressed as:

$$\frac{\partial[^{3i}SO_4]}{\partial t} = D_S \frac{\partial^2[^{3i}SO_4]}{\partial z^2} - s\frac{\partial[^{3i}SO_4]}{\partial z} - R[^{3i}SO_4] \tag{4}$$

where $[^{3i}SO_4]$ is the porewater concentration of ³⁴SO₄ and ³²SO₄, $z$ is the depth below the redox boundary of DSR, $D_s$ is the vertical diffusivity coefficient of bulk sediment (m² year⁻¹), $s$ is the sedimentation rate (cm ky⁻¹), $R$ is the first-order rate constant during DSR.

Sulfur isotopic fractionation during DSR can be regard as the different reaction rate constant between ³⁴S and ³²S. Assuming a steady state ($\frac{\partial[^{3i}SO_4]}{\partial t} = 0$, with invariant $D$, $s$, $R$), the solution of Eq. (4) is given by:

$$[^{3i}SO_4] = [^{3i}SO_4]_0 e^{\left(\frac{s - \sqrt{(s^2 + 4R_{3i}D_S)}}{2D_S}\right) \times z} \tag{5}$$

The porewater sulfur isotope ($\delta^{34}SO_{4pw}$) profile can be calculated by:

$$\delta^{34}SO_{4pw} = \left( \ln[^{34}SO_4]/[^{32}SO_4] - \ln[^{34}S]_{std}/[^{32}S]_{std} \right) \times 1000 \tag{6}$$

The subscript std denotes the standard. At any depth below the upper bound of DSR zone, the sulfur isotope of instantaneous $H_2S$ formation can be calculated by:

$$\delta^{34}S_{H_2S} = \delta^{34}SO_{4pw} - \Delta_{DSR} \tag{7}$$

where $\Delta_{DSR}$ is the biological sulfur isotope fractionation during DSR between sulfate and $H_2S$.

Pyrite precipitation within sediment is a continuous process within the DSR zone with the lower bound marked by the depletion of porewater sulfate. To calculate the instantaneous sulfur isotope of pyrite within the DSR zone, we divide sediments into different depth slices ($i$) starting from the upper bound of DSR zone ($i = 0$). Assuming no sulfur isotope fractionation during pyrite precipitation between $H_2S$ and pyrite, $\delta^{34}S_{py}$ at depth $z$ can be calculated by:

$$\delta^{34}S_{py} = \frac{\sum_0^i \left( \delta^{34}S_{H_2S} \times [SO_4]^i \right)}{\sum_0^i [SO_4]^i} \tag{8}$$

The depth $z$ of the slice $i$ can be calculated as $z = i \times h$, where $h$ is the thickness of each slice. Assuming all $H_2S$ is precipitated as pyrite, the cumulative amount of pyrite formation within DSR ($M^z$) can expressed as:

$$M^z = \sum_0^i \left( [SO_4]^i \times R \times \frac{h}{s} \right) \tag{9}$$

1D-DR model. Pyrite can precipitate in porewater with $H_2S$ diffusion from overlying water column. This process can be quantified by the one-dimensional diffusion-reaction (1D-DR) model. This model includes two processes: $H_2S$ diffusion and pyrite precipitation. The sulfide diffusion is driven by the $H_2S$ concentration gradient between sulfidic seawater and sediment porewater, which is generated by pyrite precipitation in sediment porewater. The 1D-DR model is expressed as:

$$\frac{\partial[H^{3i}S]}{\partial t} = D_s \frac{\partial^2[H^{3i}S]}{\partial z^2} - R[H^{3i}S] \tag{10}$$

Where the $[H^{3i}S]$ is the porewater concentration of $H^{34}S$ and $H^{32}S$, $z$ is the depth below the water-sediment surface, $D_s$ is the vertical diffusivity coefficient of bulk sediment (m² year⁻¹), $R$ is the first-order rate constant for pyrite formation via $H_2S$ reaction with reactive Fe.

Assuming a steady state, the solution for this equation is,

$$[H^{3i}S] = [H^{3i}S]_0 e^{\frac{-\sqrt{4R_{3i}D_s}}{2D_s} \times z} \tag{11}$$

$\delta^{34}S_{H2S}$ profile can be calculated by:

$$\delta^{34}S_{H2S} = \left(\ln[H^{34}S]/[H^{32}S] - \ln[^{34}S]_{std}/[^{32}S]_{std}\right) \times 1000 \qquad (12)$$

$[^{3i}S]_{std}$ represents the standard of $^{34}S$ or $^{32}S$. The instantaneous $\delta^{34}S_{py}$ can be calculated by:

$$\delta^{34}S_{py} = \delta^{34}S_{H2S} - \Delta_{py} \qquad (13)$$

where $\Delta_{py}$ represents the sulfur isotope fractionation during pyrite precipitation. The $\delta^{34}S_{py}$ at depth $z$ below water-sediment surface can be calculated by:

$$\delta^{34}S_z = \frac{\sum_0^i \delta^{34}S_{py} \times [HS]^i}{\sum_0^i [HS]^i} \qquad (14)$$

The cumulative amount of pyrite formation can be calculated by Eq. (9).

Sulfur isotope model for $\Delta^{33}S_{py}$–$\delta^{34}S_{py}$ plot. Assuming all the Nantuo pyrite were formed within ferruginous porewater and pyrite formation was rapid and irreversible. $\delta^{34}S_{py}$ was modeled by a sulfate concentration model[37]. In a steady state, sulfate concentration at specific depth can be expressed as:

$$SO_{4z} = (SO_{40} - SO_{4\infty}) \times e^{-\frac{k}{s}z} + SO_{4\infty} \qquad (15)$$

where, $SO_{4z}$ represents sulfate concentration at any given depth $z$, $SO_{40}$ represents the initial sulfate concentration, $SO_{4\infty}$ represent sulfate concentration at infinitive depth, i.e., sulfate left after DSR, $k$ is sulfate reduction rate constant, $s$ is the sedimentation rate.

As the little mass of $^{36}S$, here the equation of total mass of sulfate can be simplified as $SO_{4total} = {}^{32}SO_{4z} + {}^{33}SO_{4z} + {}^{34}SO_{4z}$.

For sulfur isotope, Eq. (15) can be rewritten as:

$$\delta^{3i}S_{SO_4} = \left[\frac{\left(\frac{^{3i}SO_4}{^{32}SO_4}\right)}{\left(\frac{^{3i}S}{^{32}S}\right)_{CDT}} - 1\right] \times 1000 \qquad (16)$$

Here, i equals 3 or 4.

$\delta^{3i}S$ of porewater sulfate is controlled by the initial seawater sulfate $\delta^{3i}S$, sulfate reduction rate constant $k$, sedimentation rate $s$ and sulfur isotope fractionation factor ($\alpha$) for DSR[37]. Defined $\alpha$ as the ratio of sulfate reduction rate (SRR) of $^{33}S$ or $^{34}S$ over the $^{32}S$ normalized with corresponding isotope concentration:

$$^{3i}\alpha = \frac{\left(\frac{^{3i}SRR}{^{3i}S}\right)}{\left(\frac{^{32}SRR}{^{32}S}\right)} \qquad (17)$$

Then, $\Delta^{33}S$ of porewater sulfate can be calculated:

$$\Delta^{33}S_{SO_4} = \delta^{33}S_{SO_4} - 1000 \times \left[\left(1 + \frac{\delta^{34}S_{SO_4}}{1000}\right) \times {}^{33}\lambda - 1\right] \qquad (18)$$

where, $^{33}\lambda$ is the slope of terrestrial mass fractionation line.

The instantaneous $H_2S$ sulfur isotopic compositions can be calculated by the ratios of DSR rate at corresponding depth:

$$\delta^{3i}H_2S = \left[\frac{\left(\frac{^{3i}SRR}{^{32}SRR}\right)}{\left(\frac{^{3i}S}{^{32}S}\right)_{CDT}} - 1\right] \times 1000 \qquad (19)$$

Pyrite sulfur isotopic compositions can be calculated by the accumulated $\delta^{3i}H_2S$.

**Determination of pyrite concretion content.** Pyrite concretion content was determined by using the photograph area proportion method. Pyrite concretion-bearing Nantuo diamictite and siltstone layer were photographed in field. For each field photograph, pyrite concretion content ($C_{py}$) can be calculated by:

$$C_{py} = \frac{A_{py}}{A_T} \times 100\% \qquad (20)$$

where $A_{py}$ is pyrite concretion area in the photo and $A_T$ is the total area of host rock in the photo.

The average content of pyrite concretion in each section can be calculated by:

$$C = \frac{\sum_1^n [C_{py}]^n}{n} \qquad (21)$$

where $n$ is the total number of analyzed field photographs.

**Data availability**. The data that support the findings of this study are available from the corresponding author upon reasonable request.

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

## Acknowledgments

This study is supported by the Strategic Priority Research Program (B) of Chinese Academy of Sciences (Grant number XDB18000000), Natural Science Foundation of China (41322021) and State Key Laboratory of Palaeobiology and Stratigraphy (Nanjing Institute of Geology and Paleontology, CAS) (No. 183114). P.W.C. acknowledges funding from an NSERC PGS-D fellowship, and the Agouron Institute Postdoctoral Fellow Program.

## Author contributions

B.S. designed the project. X.L., B.S., K.H. and H.M. conducted field work. X.L., A.J.K., P.W.C. and Y.P. analyzed data. X.L., W.T. and B.S. performed modeling analyses. X.L., B.S., Y.P., S.X., C.Z., H.B. and Y.L. led data interpretation and developed the manuscript. All authors contributed to discussion and manuscript revision.
