## [Peer Review File · Nature Communications]

Reviewers' comments:

Reviewer #1 (Remarks to the Author):

This paper reports sulfur contents and multiple sulfur isotopes data on Marinoan diamictite sections. This snowball Earth glacial deposit present some pyrite enriched levels composed of both nodules and aggregated pyrite levels. The author uses the data obtained to conclude on a "widespread presence of oceanic Euxinia"

The idea that the isotope values and pyrite contents may reflect the presence of Euxinia is very interesting and the model they use is very original. I thus recommend this paper for publication in Nature Communication but with moderate revisions. In general, I would argue that the authors try to make a general case while their signal could be localized to a restricted diagenetic environment. I have four major concerns and recommendations about the data and interpretation presented and some small comments:

1) The author does not seem to discuss the possible lithology effect on the isotopic and content record. They claim no lithological effect. On their log you can clearly see distinctions between massive diamictite and Gravelly siltstone nodules size and contents.

2) The proposed model for diffusion is elegant and very attractive however, the parameters used in their model are not explain and given. For example, it seems that the same vertical diffusive effect value is used for all the sections and all lithology's? It would be very helpful for the reader to have this value and know how this constant value affect the model results.

3) This paper is remarkable mostly because of the use of the pyrite content in the sediment. They clearly stated that high sedimentary pyrite content (together with the high $\delta^{34}\text{S}$ values) can be obtained only involving euxinic environment (explaining the large pool of available S). My concern is on this sedimentary pyrite. Pyrite is on the form of Nodules which is clearly diagenetic. Interestingly these nodules appear close to a fluid preferential circulation interface made by the diamictite-carbonate transition. I would clearly recommend showing some photos and thin sections of the matrice (not of the nodule) and develop the context of precipitation of these nodules. Why the H_2S could not come from lateral migration or even underlying layers below then accumulated at the lithological interface? In such case the sedimentary total pyrite content would not be so relevant.

4) One of the reagent which limit BSR activity is the available organic matter (which is only part of the total organic matter). In the case of an early diagenetic precipitation I would expect to see organic matter rich sediment nearby. How much organic matter oxidation the Nodules presence involve? Does the author suggest that this organic matter was produced during the Glaciation?

More minor comments:

Generally, the authors should use the larger space and high number of citations authorized by Nature Communications.

-Line 40: In the sentence, the authors seem to suggest that it is the high pCO_2 which imply a redox stratification. Does that mean that before the glaciation, when pCO_2 was lower the ocean was entirely oxic? Can the authors explain the relation between high pCO_2 in the end of the Snowball Earth and redox stratification?

-Line 78: On the photo figure S2F. Concretions are smaller than 5 cm which is not in agreement with the text.

-Line 91: I see a gradient in Figure S2H with the center at around -15.95 and the extremity at -17

to -19. To affirm that “there is no rim to core differences in either pyrite content or crystal size” I’d like to recommend the author to do core to rim analyses on other concretions.

-Line 98: Is the author basing this sentence on Figure S2H? In such case, please refer to Figure S2H and see my comment above.

-Line 119: In the description of the results and the Log of the section, I would not refute the lithological effect on the isotopic values. In the Open Shelf the concretions are in gravelly siltstone, compare to the basin section where the concretions are in massive diamictites. How can the authors relate this possible dependency? Please add a bit of description in the main text.

Line 120: I do not understand this argument. If the hydrothermalism comes from the deeper levels you should expect higher pyrite content in the Basinal sections, reformulate.

Line 125: Agree with the isotopic argument but it can still reflect a mix between two reservoirs which would explain too the positive C33S.

Line 133: Please homogenize DSR, BSR, MSR (in both text and figure)

Line 170: More BSR activity in the basin would reflect more organic matter availability? I think that the manuscript would benefit if the author were discussing more this aspect in regard to the Snowball conditions.

Line 185-190: In such cases, wouldn’t we expect a nutrient gradient from the cost to the deeper part of the platform? And thus, more organic matter in the shelf?

-Fig S2 photo quality not good, S2D the yellow arrow seems to indicate a facies transition and not a pyrite aggregates. To my knowledge there is no photo from the open shelf pyrites.

-Fig. 4: it is complicated to understand on the figure what processes causes the isotopic fractionation and record. For example, on Figure 4A the 46 per mil close to “diffusion” give the idea that the diffusion is creating the fractionation. The sulfate diffuses then it is the BSR which fractionate. Please clarify this.

Reviewer #2 (Remarks to the Author):

This paper documents large pyrite concretions from the top of Marinoan diamictites, and the authors argue that the stratigraphic and geochemical observations indicate that pyrite was ultimately derived from water column H₂S, rather than via sulfate reduction occurring in sediments. The argument is made on the basis that sedimentary-MSR-pyrite should be more abundant in nearshore environments with high productivity, and is developed via a 1D advection-diffusion model, in which the δ³⁴S composition of sedimentary-MSR-pyrite cannot be made to exceed ~10 per mil.

The subject matter is of course very interesting, and I generally think that the authors explanation sounds reasonable. That said, the conclusions here rely heavily on the 1D advection-diffusion modelling, but there is very little explanation of why the model produces these boundaries, what the values of the various parameters are, and their uncertainties.

Specifically, it is not noted in the main text, methods or SI why the 1D-DAR model produces the upper bound of 10 per mil. Figure 3a shows the upper bound but it is not clear why all of the lines stop at 10 per mil, unlike the Rayleigh model which clearly asymptotes. This should be explained clearly. It is also unclear how sensitive this boundary is to model parameters e.g. sulfate

concentration, sediment diffusivity. I did not find all parameters listed in the methods and SI so was unable to repeat the calculations.

The paper read well but there are some minor grammatical errors, I have tried to note these below but have probably missed several. A full proof read is advised.

In general I think this is an interesting piece of work and deserves publication, but it needs a much clearer explanation of the modelling (especially given the general readership of the journal), and a stronger defense of the model-based conclusions.

Minor points

Line 25. "followed by an overturning" should be reworded to explain more clearly what this means

Line 26. "seen" can be replaced by "saw" to read better

Line 49. "and modelling study" can be replaced by "and a modelling study"

Line 53. "of Earth system" can be replaced by "of the Earth system"

Line 56. "immediately lay beneath" can be replaced with "lie immediately beneath"

Line 56. The pyrite concretions haven't directly recorded ocean, atmosphere and biological conditions. I understand what the authors are saying here but it should be reworded.

Line 134. "in sulfidic" can be replaced with "in a sulfidic"

Line 179. "and high concentration" can be replaced with "and a high concentration"

Line 181-182. This needs to be explained in more detail. What is it about bathymetric difference that controls concretion content?

Line 186. "need to be sustained" can be replaced with "can only be sustained" or similar

Line 187. "is the major sources" can be replaced by "is the major source"

Reviewers' comments:

Reviewer #1 (Remarks to the Author):

This paper reports sulfur contents and multiple sulfur isotopes data on Marinoan diamictite sections. This snowball Earth glacial deposit present some pyrite enriched levels composed of both nodules and aggregated pyrite levels. The author uses the data obtained to conclude on a “widespread presence of oceanic Euxinia”

The idea that the isotope values and pyrite contents may reflect the presence of Euxinia is very interesting and the model they use is very original. I thus recommend this paper for publication in Nature Communication but with moderate revisions. In general, I would argue that the authors try to make a general case while their signal could be localized to a restricted diagenetic environment. I have four major concerns and recommendations about the data and interpretation presented and some small comments:

- 1) The author does not seem to discuss the possible lithology effect on the isotopic and content record. They claim no lithological effect. On their log you can clearly see distinctions between massive diamictite and Gravelly siltstone nodules size and contents.

Response: It is common that a gravelly siltstone layer normally overlies the massive diamictite at the top of Nantuo Formation. Pyrite concretions could be found in both lithologies, and the sulfur isotopes and pyrite contents can be different in the two lithologies in some localities. In the slope sections, gravelly siltstones normally contain disseminated pyrites rather than pyrite concretions, and the disseminated pyrites have wider range of variation in $\delta^{34}\text{S}_{\text{py}}$. In contrast, small pyrite nodules could be found in the gravelly siltstone with significantly lower $\delta^{34}\text{S}_{\text{py}}$ in the shelf sections. We clarified this issue in the revised manuscript.

Lines 141-146: $\delta^{34}\text{S}$ values of disseminated pyrite extracted from gravelly siltstone in the top Nantuo Formation shows greater fluctuation (13.7‰ to 48.1‰, mean = 28.9‰) in the Taoying section but limited variation (-5.0‰ to -5.7‰, mean = -5.4‰) in the Bahuang section. In the outer shelf sections, pyrite concretions from the gravelly siltstone layers have the lowest $\delta^{34}\text{S}_{\text{py}}$ values [Songlin (8.1‰ to 12.6‰, mean = 11.1‰), Youxi (-5.5‰ to -2.5‰, mean = -4.0‰)].

Lines 159-161: Pyrite concretions can be observed both in gravelly siltstone and massive diamictite in the slop environment, whereas in the shallow water facies pyrite nodules only occur in gravelly siltstone.

Lines 222-227: Because $\delta^{34}\text{S}_{\text{py}}$ mirrors the isotopic composition of H_2S in seawater³³, the variation in $\delta^{34}\text{S}_{\text{py}}$ reflects the isotopic heterogeneity of seawater H_2S . The relatively consistent $\delta^{34}\text{S}_{\text{py}}$ values observed in the basal samples suggest that the H_2S concentration was more or less homogenous in most distal seawaters. For the same reason, the more variable $\delta^{34}\text{S}_{\text{py}}$ values of the slope sections reflect a rapid oscillation of seawater H_2S concentrations.

- 2) The proposed model for diffusion is elegant and very attractive however, the parameters used in their model are not explain and given. For example, it seems that the same vertical diffusive effect value is used for all the sections and all lithologies? It would be very helpful for the reader to have this value and know how this constant value affect the model results.

Response: We adopted reviewer's recommendation. In revision, (1) we clarify each default parameters in the supplementary material (Supplementary Note 2, and supplementary Table 3-5), (2) we conduct sensitivity test for each unconstrained parameters (Supplementary Figs. 6-8).

- 3) This paper is remarkable mostly because of the use of the pyrite content in the sediment. They clearly stated that high sedimentary pyrite content (together with the high $\delta^{34}\text{S}$ values) can be obtained only involving euxinic environment (explaining the large pool of available S). My concern is on this sedimentary pyrite. Pyrite is on the form of Nodules which is clearly diagenetic. Interestingly these nodules appear close to a fluid preferential circulation interface made by the diamictite-carbonate transition. I would clearly recommend showing some photos and thin sections of the matrix (not of the nodule) and develop the context of precipitation of these nodules. Why the H_2S could not come from lateral migration or even underlying layers below then accumulated at the lithological interface? In such case the sedimentary total pyrite content would not be so relevant.

Response: The Nantuo pyrite is early diagenetic, i.e. authigenic. Multiple lines of evidence show that these pyrite concretions were formed within porewater during early-stage diagenesis instead of a late-stage diagenetic origin or hydrothermal origin. We clarified this point in the revised manuscript (Lines 165-167, 174-180). In addition to the evidence proposed in the manuscript, we also claim that if H_2S came from lateral migration, the pyrite concretions both from shallow water facies and deep water facies should have same $\delta^{34}\text{S}_{\text{py}}$ values, which is inconsistent with the observation. To further support our argument, as suggested by reviewer, we add 2 photographs of the matrix (Supplementary Fig. 3g and 3h).

Lines 165-167: Pyrite nodules are preserved in the matrix of cap carbonate and, predating the early diagenetic sheet crack structures and arguing against a late diagenetic or hydrothermal origin.

Lines 174-180: Petrological evidence suggest that the Nantuo pyrite concretions were formed in sediment during early diagenesis. Pyrite crystals are tightly packed with interstitial spaces filled with clasts or cemented by silica, or float within a siliciclastic matrix, suggesting early concretion formation before sediment compaction (Supplementary Figs. 2d, 3a and 3b). Furthermore, the matrix of pyrite-bearing diamictite and gravelly siltstone also contains some disseminated euhedral pyrite, (Supplementary Fig. 3g and 3h), suggesting an authigenic origin.

- 4) One of the reagent which limit BSR activity is the available organic matter (which is

only part of the total organic matter). In the case of an early diagenetic precipitation I would expect to see organic matter rich sediment nearby. How much organic matter oxidation the Nodules presence involve? Does the author suggest that this organic matter was produced during the Glaciation?

Response: BSR needs to be sustained by the availability of organic matter. This prompted us to conclude that the recovery of primary productivity was the primary driver of the oceanic euxinia. Organic rich diamictite, e.g. dark-grey, layered diamictites shown in Supplementary Fig. 2e at the top of the Nantuo Formation is evident. However, we would stress that organic matter content may not necessarily be correlated with pyrite content, because sulfate reduction was occurring in water column instead of in porewater. In the process, most organic matter would be degraded in water column. We also stress that organic matter was produced during or after deglaciation, rather than produced in glaciation. We clarified these issues in the revised manuscript.

Lines 242-250: We hypothesize that, during the deglaciation of the Marinoan glaciation, enhanced continental weathering²⁰ could have delivered abundant nutrients and sulfate into the ocean. Because the deglacial continental weathering may predate oceanic euxinia²⁰ due to the delayed recovery of productivity in acidified seawater, both nutrients and sulfate might be accumulated in seawater from riverine inputs. Once surface ocean productivity was resumed at high levels, high nutrient content could sustain high productivity, while DSR in water column would be fueled by sufficient supplies of organic matter and sulfate, driving oceanic euxinia.

More minor comments:

Generally, the authors should use the larger space and high number of citations authorized by Nature Communications.

Response: We have added additional 14 citations in the revision, such as Lyons et al. 2009, 2010; Poulton et al. 2011; Hurtgen et al. 2006 and others...

-Line 40: In the sentence, the authors seem to suggest that it is the high pCO₂ which imply a redox stratification. Does that mean that before the glaciation, when pCO₂ was lower the ocean was entirely oxic? Can the authors explain the relation between high pCO₂ in the end of the Snowball Earth and redox stratification?

Response: We are sorry that we did not make a clear statement in the original manuscript. In the revised manuscript, we have added the following additional information about the relationship between pCO₂ and redox stratification in the revision.

Lines 59-63: "Such a high pCO₂ atmosphere will result in a rapid meltdown of glaciers¹⁰ and an extreme greenhouse climate^{6,7}. Rapid deglaciation may lead to the shutdown of thermohaline circulation and the intensification of ocean stratification¹¹, with warmer and dysoxic freshwater in surface ocean overlying cold and anoxic seawater in the deep^{10,12}."

-Line 78: On the photo figure S2F. Concretions are smaller than 5 cm which is not in agreement with the text.

Response: The reported 5-15 cm nodules in the slope section are the statistical argument, suggesting most nodules are within this range. However, there are certainly other nodules out of this size range.

-Line 91: I see a gradient in Figure S2H with the center at around -15.95 and the extremity at -17 to -19. To affirm that “there is no rim to core differences in either pyrite content or crystal size” I’d like to recommend the author to do core to rim analyses on other concretions.

Response: Yes, there is some minor variations in sulfur isotopes from the rim to core. See our response for Line 98 below.

-Line 98: Is the author basing this sentence on Figure S2H? In such case, please refer to Figure S2H and see my comment above.

Response: Yes, we agree with reviewer that there is a subtle gradient in $\delta^{34}\text{S}$ from the core to rim as shown in Fig. S2h. Indeed, we have analyzed 17 pyrite concretions collected from the basin outcrops, and each pyrite display a minor variation in the $\delta^{34}\text{S}$ from the core to the rim (differences mostly <5‰), while not all pyrite nodules display a clear rim to core gradient. In many pyrite nodules, the spatial distribution of $\delta^{34}\text{S}$ is random.

-Line 119: In the description of the results and the Log of the section, I would not refute the lithological effect on the isotopic values. In the Open Shelf the concretions are in gravelly siltstone, compare to the basin section where the concretions are in massive diamictites. How can the authors relate this possible dependency? Please add a bit of description in the main text.

Response: In the revised manuscript, distribution and sulfur isotopic compositions of pyrites in different lithologies are differentiated. We clearly state that the pyrite from different lithologies are different, e.g. disseminated vs. nodules, and the variation of $\delta^{34}\text{S}$.

Line 120: I do not understand this argument. If the hydrothermalism comes from the deeper levels you should expect higher pyrite content in the Basinal sections, reformulate.

Response: We agree with reviewer that our original statement was simplified by assuming that the migration of hydrothermal fluid was faster than the precipitation of pyrite. In this case, accumulation of hydrothermal fluids in topographic highs would result in higher pyrite content in the shallow water settings. Alternatively, fast pyrite formation and/or slow migration of hydrothermal fluids would results in high pyrite contents in the basinal sections. Therefore, we remove this point in the revision.

Line 125: Agree with the isotopic argument but it can still reflect a mix between two reservoirs which would explain too the positive C33S.

Response: Yes, we agree with reviewer that mixing of two or several reservoirs could explain the positive $\Delta^{33}\text{S}$ data. However, since our data were derived from 7 sections spanning from shelf to basinal sections, mixing may not the only reason for the positive $\Delta^{33}\text{S}$ data.

Line 133: Please homogenize DSR, BSR, MSR (in both text and figure)

Response: In the revised manuscript, we only use DSR, and BSR and MSR are no longer

used.

Line 170: More BSR activity in the basin would reflect more organic matter availability? I think that the manuscript would benefit if the author were discussing more this aspect in regard to the Snowball conditions.

Response: More pyrite in the basin may reflect more H₂S supply from sulfidic seawater, which may result from (1) higher concentration of H₂S in water column, and (2) the greater thickness of sulphidic water mass. The decrease of concretion density and content from the basin to the shallow water facies is consistent with the onshore-offshore bathymetric gradient. Although variation of organic matter may also cause fluctuation of H₂S concentration in water column, however, there is no direct evidence indicating such differences.

We suggest that, in the context of snowball Earth event, the deglaciation predated cap carbonate precipitation and recovery of primary productivity, resulting in the accumulation and homogenization of nutrient and sulfate in the water column. In such case, water depth would be the primary control on organic matter production and BSR activity.

Lines 233-236: The decrease of pyrite content from deep to shallow settings is attributed to the bathymetric differences of the depositional environments³⁸. Deeper water in the basin may have a thicker sulfidic water layer, resulting in more pyrite precipitation.

Lines 342-250: We hypothesize that, during the deglaciation of the Marinoan glaciation, enhanced continental weathering²⁰ could have delivered abundant nutrients and sulfate into the ocean. Because the deglacial continental weathering may predate oceanic euxinia²⁰ due to the delayed recovery of productivity in acidified seawater, both nutrients and sulfate might be accumulated in seawater from riverine inputs. Once surface ocean productivity was resumed at high levels, high nutrient content could sustain high productivity, while DSR in water column would be fueled by sufficient supplies of organic matter and sulfate, driving oceanic euxinia.

Line 185-190: In such cases, wouldn't we expect a nutrient gradient from the coast to the deeper part of the platform? And thus, more organic matter in the shelf?

Response: Yes, if the riverine input was the major sources of nutrients supply, it is reasonable to expect that a nutrient gradient would exist from shallow water to the deep. However, in the context of snowball Earth condition, the deglacial continental weathering might predate recovery of primary productivity, allowing the accumulation and homogenization of nutrient in the water column, and eliminating the nutrient gradient between shallow and deep water settings. In the recovery of primary productivity, shallow water depth (10s of meters) in the platform settings would result in less organic matter production per unit area than in the deep water settings (100s of meters). Low organic matter production in shallow water settings is consistent with generally low $\delta^{34}\text{S}_{\text{py}}$ in the shelf settings.

-Fig S2 photo quality not good, S2D the yellow arrow seems to indicate a facies transition and not a pyrite aggregates. To my knowledge there is no photo from the open shelf pyrites.

Response: We have replaced the S2d with a high quality photo and also have added a new photo of pyrite from the open shelf (Supplementary Fig. 2g).

-Fig. 4: it is complicated to understand on the figure what processes causes the isotopic fractionation and record. For example, on Figure 4A the 46 per mil close to “diffusion” give the idea that the diffusion is creating the fractionation. The sulfate diffuses then it is the BSR which fractionate. Please clarify this.

Response: - We adopted reviewer’s recommendation. In the revision, we put the fractionation factor close to “DSR”.

Reviewer #2 (Remarks to the Author):

This paper documents large pyrite concretions from the top of Marinoan diamictites, and the authors argue that the stratigraphic and geochemical observations indicate that pyrite was ultimately derived from water column H₂S, rather than via sulfate reduction occurring in sediments. The argument is made on the basis that sedimentary-MSR-pyrite should be more abundant in nearshore environments with high productivity, and is developed via a 1D advection-diffusion model, in which the $\delta^{34}\text{S}$ composition of sedimentary-MSR-pyrite cannot be made to exceed ~10 per mil.

The subject matter is of course very interesting, and I generally think that the authors explanation sounds reasonable. That said, the conclusions here rely heavily on the 1D advection-diffusion modelling, but there is very little explanation of why the model produces these boundaries, what the values of the various parameters are, and their uncertainties.

Specifically, it is not noted in the main text, methods or SI why the 1D-DAR model produces the upper bound of 10 per mil. Figure 3a shows the upper bound but it is not clear why all of the lines stop at 10 per mil, unlike the Rayleigh model which clearly asymptotes. This should be explained clearly. It is also unclear how sensitive this boundary is to model parameters e.g. sulfate concentration, sediment diffusivity. I did not find all parameters listed in the methods and SI so was unable to repeat the calculations.

The paper read well but there are some minor grammatical errors, I have tried to note these below but have probably missed several. A full proof read is advised.

In general I think this is an interesting piece of work and deserves publication, but it needs a much clearer explanation of the modelling (especially given the general readership of the journal), and a stronger defense of the model-based conclusions.

Response: Our original model was based on the assumption of unlimited supply of organic matter, i.e. pyrite production was solely controlled by the availability of sulfate. Thus, the calculated pyrite sulfur isotope of ~10‰ represents the maximum estimation (Supplementary Figs. 6–9). If organic matter supply was limited, pyrite's $\delta^{34}\text{S}$ would be lower than 10‰. In the supplementary material, we expanded the model, in which the availability of organic matter is considered (Supplementary Fig. 11). Unlike the original equation that has an analytical solution, the new model only has numerical solutions.

In addition, we have run the sensitivity tests of the model, showing how the model results ($\delta^{34}\text{S}_{\text{py}}$ and pyrite content) are effected by those loosely constrained factors, including sedimentation rate, diffusivity coefficient, seawater sulfate concentration and sulfate reduction rate constant (Supplementary Figs. 6–9). The sensitivity tests indicate that our conclusion based on the $\delta^{34}\text{S}_{\text{py}}$ -pyrite content relationship is not affected by the selection of parameters.

Minor points

Response: We have corrected the grammatical errors the reviewer so kindly pointed out for

us. Jay Kaufman, one of our co-authors, has personally taken the responsibility for English grammar errors still left in the text.

Line 25. “followed by an overturning” should be reworded to explain more clearly what this means

Line 26. “seen” can be replaced by “saw” to read better

Line 49. “and modelling study” can be replaced by “and a modelling study”

Line 53. “of Earth system” can be replaced by “of the Earth system”

Line 56. “immediately lay beneath” can be replaced with “lie immediately beneath”

Line 56. The pyrite concretions haven’t directly recorded ocean, atmosphere and biological conditions. I understand what the authors are saying here but it should be reworded.

Line 134. “in sulfidic” can be replaced with “in a sulfidic”

Line 179. “and high concentration” can be replaced with “and a high concentration”

Line 181-182. This needs to be explained in more detail. What is it about bathymetric difference that controls concretion content?

Line 186. “need to be sustained” can be replaced with “can only be sustained” or similar

Line 187. “is the major sources” can be replaced by “is the major source”

REVIEWERS' COMMENTS:

Reviewer #1 (Remarks to the Author):

Lang et al. made a nice effort in reviewing the paper and have considered most of my recommendations. I now recommend the publication of the paper in Nature communication.

Reviewer #2 (Remarks to the Author):

The revision of this paper has made the modeling much clearer. The authors have added useful plots to the SI which help the reader to better understand where the summary plots in the main text come from and how the model responds to various uncertainties. I am happy with the authors' conclusions and support publication of the paper with a minor reservation:

The reader is currently required to go to the SI to have much of a chance at fully understanding figure 3. This is because figure 3 plots model 'phase states' (one dependent variable against another) and does not include any of the plots against depth. I would recommend including at least one model plot versus depth in the main text so the work can be understood in summary without having to look in the SI. I am happy for the authors and editor to agree on this and do not need to see the paper again.

REVIEWERS' COMMENTS:

Reviewer #1 (Remarks to the Author):

Lang et al. made a nice effort in reviewing the paper and have considered most of my recommendations. I now recommend the publication of the paper in Nature communication.

Reviewer #2 (Remarks to the Author):

The revision of this paper has made the modeling much clearer. The authors have added useful plots to the SI which help the reader to better understand where the summary plots in the main text come from and how the model responds to various uncertainties. I am happy with the authors' conclusions and support publication of the paper with a minor reservation:

The reader is currently required to go to the SI to have much of a chance at fully understanding figure 3. This is because figure 3 plots model 'phase states' (one dependent variable against another) and does not include any of the plots against depth. I would recommend including at least one model plot versus depth in the main text so the work can be understood in summary without having to look in the SI. I am happy for the authors and editor to agree on this and do not need to see the paper again.

Response: Thanks for reviewer's great suggestion. We have added 2 model plots versus depth in figure 3 in the revision.